# INTEGRATING CATEGORICAL SEMANTICS INTO UNSUPERVISED DOMAIN TRANSLATION

**Samuel Lavoie**[*]**, Faruk Ahmed & Aaron Courville**[†]
Département d'Informatique et de Recherche Opérationnelle
Université de Montréal, Mila

## ABSTRACT

While unsupervised domain translation (UDT) has seen a lot of success recently, we argue that mediating its translation via categorical semantic features could broaden its applicability. In particular, we demonstrate that categorical semantics improves the translation between perceptually different domains sharing multiple object categories. We propose a method to learn, in an unsupervised manner, categorical semantic features (such as object labels) that are invariant of the source and target domains. We show that conditioning the style encoder of unsupervised domain translation methods on the learned categorical semantics leads to a translation preserving the digits on MNIST↔SVHN and to a more realistic stylization on Sketches→Reals.[1]

## 1 INTRODUCTION

Domain translation has sparked a lot of interest in the computer vision community following the work of Isola et al. (2016) on *image-to-image translation*. This was done by learning a conditional GAN (Mirza & Osindero, 2014), in a supervised manner, using paired samples from the source and target domains. CycleGAN (Zhu et al., 2017a) considered the task of unpaired and unsupervised image-to-image translation, showing that such a translation was possible by simply learning a mapping and its inverse under a *cycle-consistency* constraint, with GAN losses for each domain.

But, as has been noted, despite the cycle-consistency constraint, the proposed translation problem is fundamentally ill-posed and can consequently result in arbitrary mappings (Benaim et al., 2018; Galanti et al., 2018; de Bézenac et al., 2019). Nevertheless, CycleGAN and its derivatives have shown impressive empirical results on a variety of image translation tasks. Galanti et al. (2018) and de Bézenac et al. (2019) argue that CycleGAN's success is owed, for the most part, to architectural choices that induce implicit biases toward *minimal complexity* mappings. That being said, CycleGAN, and follow-up works on unsupervised domain translation, have commonly been applied on domains in which a translation entails little geometric changes and the style of the generated sample is independent of the semantic content in the source sample. Commonly showcased examples include translating edges↔shoes and horses↔zebras.

While these approaches are not without applications, we demonstrate two situations where unsupervised domain translation methods are currently lacking. The first one, which we call *Semantic-Preserving Unsupervised Domain Translation* (SPUDT), is defined as translating, without supervision, between domains that share common semantic attributes. Such attributes may be a non-trivial composition of features obfuscated by domain-dependent spurious features, making it hard for the current methods to translate the samples while preserving the shared semantics despite the implicit bias. Translating between MNIST↔SVHN is an example of translation where the shared semantics, the digit identity, is obfuscated by many spurious features, such as colours and background distractors, in the SVHN domains. In section 4.1, we take this specific example and demonstrate that using domain invariant categorical semantics improves the digit preservation in UDT.

The second situation that we consider is *Style-Heterogeneous Domain Translation* (SHDT). SHDT refers to a translation in which the target domain includes many semantic categories, with a distinct

---

[*]Correspondence to: samuel.lavoie.m@gmail.com.
[†]CIFAR fellow

[1]The public code can be found: https://github.com/lavoiems/Cats-UDT

style per semantic category. We demonstrate that, in this situation, the style encoder must be conditioned on the shared semantics to generate a style consistent with the semantics of the given source image. In Section 4.2, we consider an example of this problem where we translate an ensemble of sketches, with different objects among them, to real images.

In this paper, we explore both the SPUDT and SHDT settings. In particular, we demonstrate how domain invariant categorical semantics can improve translation in these settings. Existing works (Hoffman et al., 2018; Bousmalis et al., 2017) have considered semi-supervised variants by training a classifier with labels on the source domain. But, differently from them, we show that it is possible to perform well at both kinds of tasks *without any supervision*, simply with access to unlabelled samples from the two domains. This additional constraint may further enable applications of domain translation in situations where labelled data is absent or scarce.

To tackle these problems, we propose a method which we refer to as Categorical Semantics Unsupervised Domain Translation (CatS-UDT). CatS-UDT consists of two steps: (1) learning an inference model of the shared categorical semantics across the domains of interest without supervision and (2) using a domain translation model in which we condition the style generation by inferring the learned semantics of the source sample using the model learned at the previous step. We depict the first step in Figure 1b and the second in Figure 2.

More specifically, the contributions of this work are the following:

- Novel framework for learning invariant categorical semantics across domains (Section 3.1).

- Introduction of a method of semantic style modulation to make SHDT generations more consistent (Section 3.2).

- Comparison with UDT baselines on SPUDT and SHDT highlighting their existing challenges and demonstrating the relevance of our incorporating semantics into UDT (Section 4).

## 2  RELATED WORKS

Domain translation is concerned with translating samples from a source domain to a target domain. In general, we categorize a translation that uses pairing or supervision through labels as *supervised domain translation* and a translation that does not use pairing or labels as *unsupervised domain translation*.

**Supervised domain translation** methods have generally achieved success through either the use of pairing or the use of supervised labels. Methods that leverage the use of category labels include Taigman et al. (2017); Hoffman et al. (2018); Bousmalis et al. (2017). The differences between these approaches lie in particular architectural choices and auxiliary objectives for training the translation network. Alternatively, Isola et al. (2016); Gonzalez-Garcia et al. (2018); Wang et al. (2018; 2019); Zhang et al. (2020) leverage paired samples as a signal to guide the translation. Also, some works propose to leverage a segmentation mask (Tomei et al., 2019; Roy et al., 2019; Mo et al., 2019). Another strategy is to use the representation of a pre-trained network as semantic information (Ma et al., 2019; Wang et al., 2019; Wu et al., 2019; Zhang et al., 2020). Such a representation typically comes from the intermediate layer of a VGG (Liu & Deng, 2015) network pre-trained with labelled ImageNET (Deng et al., 2009). Conversely to our work, (Murez et al., 2018) propose to use image-to-image translation to regularize domain adaptation.

**Unsupervised domain translation** considers the task of domain translation without any supervision, whether through labels or pairing of images across domains. CycleGAN (Zhu et al., 2017a) proposed to learn a mapping and its inverse constrained with a *cycle-consistency* loss. The authors demonstrated that CycleGAN works surprisingly well for some translation problems. Later works have improved this class of models (Liu et al., 2017; Kim et al., 2017; Almahairi et al., 2018; Huang et al., 2018; Choi et al., 2017; 2019; Press et al., 2019), enabling multi-modal and more diverse generations. But, as shown in Galanti et al. (2018), the success of these methods is mostly due to architectural constraints and regularizers that implicitly bias the translation toward mappings with minimum complexity. We recognize the usefulness of this inductive bias for preserving low-level features like the pose of the source image. This observation motivates the method proposed in Section 3.2 for conditioning the style using the semantics.

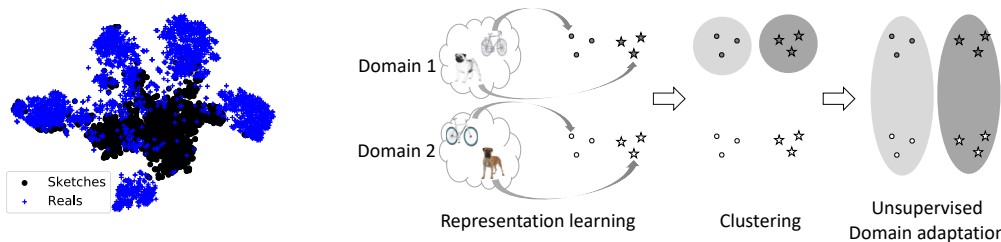

(a) ResNET-50 trained using MOCO.    (b) Domain invariant categorial representation learning.

Figure 1: (a) T-SNE embeddings of the representation of Sketches and Reals taken from a hidden layer for a pre-trained model on ImageNET, (b) Sketch of our method for learning a domain invariant categorial semantics.

## 3  CATEGORICAL SEMANTICS UNSUPERVISED DOMAIN TRANSLATION

In this section, we present our two main technical contributions. First, we discuss an unsupervised approach for learning categorical semantics that is invariant across domains. Next, we incorporate the learned categorical semantics into the domain translation pipeline by conditioning the style generation on the learned categorical-code.

### 3.1  UNSUPERVISED LEARNING OF DOMAIN INVARIANT CATEGORICAL SEMANTICS

The framework for learning the domain invariant categorical representation, summarized in Figure 1b, is composed of three constituents: unsupervised representation learning, clustering and domain adaptation. First, embed the data of the source and target domains into a representation that lends itself to clustering. This step can be ignored if the raw data is already in a form that can easily be clustered. Second, cluster the embedding of one of the domains. Third, use the learned clusters as the ground truth label in an unsupervised domain adaptation method. We provide a background of each of the constituents in Appendix A and concrete examples in Section 4. Here, we motivate their utilities and describe how they are used in the present framework.

**Representation learning.** Pre-trained supervised representations have been used in many instances as a way to preserve alignment in domain translation (Ma et al., 2019; Wang et al., 2019). In contrast to prior works that use models trained with supervision, we use models trained with self-supervision (van den Oord et al., 2018a; Hjelm et al., 2019; He et al., 2020; Chen et al., 2020a). Self-supervision defines objectives that depends only on the intrinsic information within data. This allows for the use of unlabelled data, which in turn could enable the applicability of domain translation to modalities or domains where labelled data is scarce. In this work, we consider the noise contrastive estimation (van den Oord et al., 2018b) which minimizes the distance in a normalized representation space between an anchor sample and its transformation and maximizes the distance between the same anchor sample and another sample in the data distribution. Formally, we learn the embedding function $d : \mathcal{X} \to \mathbb{R}^D$ of samples $\boldsymbol{x} \in \mathcal{X}$ as follows:

$$\arg\min_{d} -\mathbb{E}_{\boldsymbol{x}_i \sim \mathcal{X}} \log \frac{\exp(d(\boldsymbol{x}_i) \cdot d(\boldsymbol{x}_i')/\tau)}{\sum_{j=0}^{K} \exp(d(\boldsymbol{x}_i) \cdot d(\boldsymbol{x}_j)/\tau)}, \tag{1}$$

where $\tau > 0$ is a hyper-parameter, $\boldsymbol{x}_i$ is the anchor sample with its transformation $\boldsymbol{x}_i' = t(\boldsymbol{x}_i)$ and $t : \mathcal{X} \to \mathcal{X}$ defines the set of transformations that we want our embedding space to be invariant to.

While other works use the learned representation directly in the domain translation model, we propose to use it as a leverage to obtain a categorical and domain invariant embedding as described next. In some instances, the data representation is already amenable to clustering. In those cases, this step of representation learning can be ignored.

**Clustering** allows us to learn a categorical representation of our data without supervision. Some advantages of using such a representation are as follows:

- A categorical representation provides a way to select exemplars without supervision by simply selecting an exemplar from the same categorical distribution of the source sample.

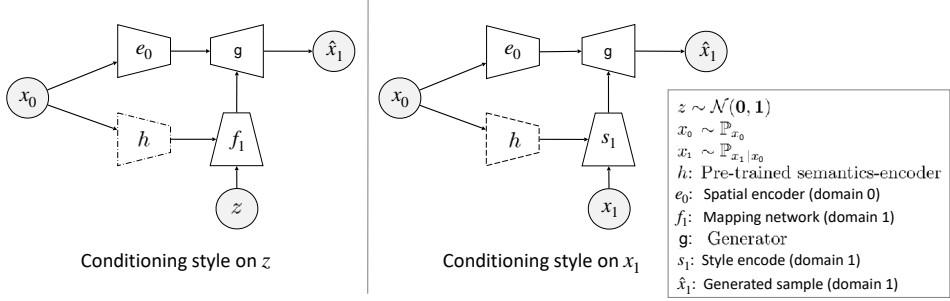

Figure 2: Our proposed adaptation to the image-to-image framework for CatS-UDT. **Left**: generate the style using a mapping network conditioned on both noise $z \sim \mathcal{N}(0, 1)$ and the semantics of the source sample $h(\boldsymbol{x}_0)$. **Right**: infer style of an exemplar $\boldsymbol{x}_2$ using a style encoder and $h(\boldsymbol{x}_0)$.

- The representation is straightforward to evaluate and to interpret. Samples with the same semantic attributes should have the same cluster.

In practice, we cluster one domain because, as we see in Figure 1a, the continuous embedding of each domain obtained from a learned model may be disjoint when they are sufficiently different. Therefore, a clustering algorithm would segregate each domain into its own clusters. Also, the domain used to determine the initial clusters is important as some domains may be more amenable to clustering than others. Deciding which domain to cluster depends on the data and the choice should be made after evaluation of the clusters or inspection of the data.

More formally, consider $\mathcal{X}_0 \subset \mathbb{R}^N$ be the domain chosen to be clustered. Assume a given embedding function $d : \mathcal{X}_0 \to \mathbb{R}^D$ that can be learned using self-supervision. If $\mathcal{X}_0$ is already cluster-able, $d$ can be the identity function. Let $c : \mathbb{R}^D \to \mathcal{C}$ be a mapping from the embedding of $\mathcal{X}_0$ to the space of clusters $\mathcal{C}$. We propose to cluster the embedding representation of the data:

$$\arg\min_c \mathrm{C}(c, d(\mathcal{X}_0)), \tag{2}$$

where $\mathrm{C}$ is a clustering objective. The framework is agnostic to the clustering algorithm used. In our experiments (Section 4), we considered IMSAT (Hu et al., 2017) for clustering MNIST and Spectral Clustering (Donath & Hoffman, 1973) for clustering the learned embedding of our real images. We give a more thorough background of IMSAT in Appendix B.3 and refer to Luxburg (2007) for a background of Spectral clustering.

**Unsupervised domain adaptation.** Given clusters learned using samples from a domain $\mathcal{X}_0$, it is unlikely that such clusters will generalize to samples from a different domain with a considerable shift. This can be observed in Figure 1a where, if we clustered the samples of the real images, it is not clear that the samples from the Sketches domain would semantically cluster as we expect. That is, samples with the same semantic category may not be grouped in the same cluster.

Unsupervised domain adaptation (Ben-David et al., 2010) tries to solve this problem where one has one supervised domain. However, rather than using labels obtained through supervision from a source domain, we propose to use the learned clusters as ground-truth labels on the source domain. This modification allows us to adapt and make the clusters learned on one domain invariant to the other domain.

More formally, given two spaces $\mathcal{X}_0 \in \mathbb{R}^N$, $\mathcal{X}_1 \in \mathbb{R}^N$ representing the data space of domains 0 and 1 respectively, given a $C$-way one-hot mapping of the embedding of domain 0 to clusters, $c : d(\mathcal{X}_0) \to \mathcal{C}$ ($\mathcal{C} \subset \{0, 1\}^C$), we propose to learn an adapted clustering $h : \mathcal{X}_0 \cup \mathcal{X}_1 \to \mathcal{C}$. We do so by optimizing:

$$\arg\min_h -\mathbb{E}_{\boldsymbol{x}_0 \sim \mathcal{X}_0} c(d(\boldsymbol{x}_0)) \log h(\boldsymbol{x}_0) + \Omega(h, \mathcal{X}_0, \mathcal{X}_1). \tag{3}$$

$\Omega$ represents the regularizers used in unsupervised domain adaptation. The framework is also agnostic to the regularizers used in practice. In our experiments, the regularizers comprised of gradient reversal (Ganin et al., 2016), VADA (Shu et al., 2018) and VMT (Mao et al., 2019). We describe those regularizers in more detail in Appendix B.5.

## 3.2 CONDITIONING THE STYLE ENCODER OF UNSUPERVISED DOMAIN TRANSLATION

Recent methods for unsupervised image-to-image translation have two particular assets: (1) they can work with few training examples, and (2) they can preserve spatial coherence such as pose. With that in mind, our proposition to incorporate semantics into UDT, as depicted in figure 2, is to incorporate semantic-conditioning into the style inference of a domain translation framework. We will consider that the semantics is given by a network ($h$ in Figure 2). The rationale behind this proposition originates from the conclusions by Galanti et al. (2018); de Bézenac et al. (2019) that the unsupervised domain translation methods work due to an inductive bias toward minimum complexity mappings. By conditioning only the style encoder on the semantics, we preserve the same inductive bias in the spatial encoder, forcing the generated sample to preserve some spatial attributes of the source sample, such as pose, while conditioning its style on the semantics of the source sample. In practice, we can learn the domain invariant categorical semantics, without supervision, using the method described in the previous subsection.

There can be multiple ways for incorporating the style into the translation framework. In this work, we follow an approach similar to the one used in StyleGAN (Karras et al., 2019) and StarGAN-V2 (Choi et al., 2019). We incorporate the style, conditioned on the semantics, by modulating the latent feature maps of the generator using an Adaptive Instance Norm (AdaIN) module (Huang & Belongie, 2017). Next, we describe each network used in our domain translation model and the training of the domain translation network.

### 3.2.1 NETWORKS AND THEIR FUNCTIONS

**Content encoders**, denoted $e$, extract the spatial content of an image. It does so by encoding an image, down-sampling it to a representation of resolution smaller or equal than the initial image, but greater than one to preserve spatial coherence.

**Semantics encoder**, denoted $h$, extracts semantic information defined as a categorical label. In our experiments, the semantics encoder is a pre-trained network.

**Mapping networks**, denoted $f$, encode $\boldsymbol{z} \sim \mathcal{N}(0, 1)$ and the semantics of the source image to a vector representing the style. This vector is used to condition the AdaIN module used in the generator which modulates the style of the target image.

**Style encoders**, denoted $s$, extract the style of an exemplar image in the target domain. This style is then used to modulate the feature maps of the generator using AdaIN.

**Generator**, denoted $g$, generates an image in the target domain given the content and the style. The generator upsamples the content, injecting the style by modulating each layer using an AdaIN module.

### 3.2.2 TRAINING

Let $\boldsymbol{x}_0 \sim \mathbb{P}_{x_0}$ and $\boldsymbol{x}_1 \sim \mathbb{P}_{x_1}$ be samples from two probability distributions on the spaces of our two domains of interest. Let $\boldsymbol{z} \sim \mathcal{N}(0, 1)$ samples from a Gaussian distribution. Let $y \sim \mathcal{B}(0.5)$ defines the domain, sampled from a Bernoulli distribution, and its inverse $\bar{y} := 1 - y$. We define the following objectives for samples generated with the mapping networks $f$ and the style encoder $s$:

**Adversarial loss** (Goodfellow et al., 2014). Constrain the translation network to generate samples in distribution to the domains. Consider $d$: the discriminators [2].

$$\mathcal{L}_{\text{adv}}^f := \mathbb{E}_y \left[ \mathbb{E}_{\boldsymbol{x}_{\bar{y}}} \log d_{\bar{y}}^f(x_{\bar{y}}) + \mathbb{E}_{\boldsymbol{x}_y} \mathbb{E}_{\boldsymbol{z}} \log(1 - d_{\bar{y}}^f(g(e_y(\boldsymbol{x}_y), f_{\bar{y}}(h(\boldsymbol{x}_y), \boldsymbol{z})))) \right],$$

$$\mathcal{L}_{\text{adv}}^s := \mathbb{E}_y \left[ \mathbb{E}_{\boldsymbol{x}_{\bar{y}}} \log d_{\bar{y}}^s(x_{\bar{y}}) + \mathbb{E}_{\boldsymbol{x}_y} \mathbb{E}_{\boldsymbol{x}_{\bar{y}} \sim \mathbb{P}_{x_{\bar{y}}|h(x_y)}} \log(1 - d_{\bar{y}}^s(g(e_y(\boldsymbol{x}_y), s_{\bar{y}}(h(\boldsymbol{x}_y), \boldsymbol{x}_{\bar{y}})))) \right]. \tag{4}$$

**Cycle-consistency loss** (Zhu et al., 2017a). Regularizes the content encoder and the generator by enforcing the translation network to reconstruct the source sample.

$$\mathcal{L}_{\text{cyc}}^f := \mathbb{E}_y \left[ \mathbb{E}_{\boldsymbol{x}_y} \mathbb{E}_{\boldsymbol{z}} \left| \boldsymbol{x}_y - g(e_{\bar{y}}(g(e_y(\boldsymbol{x}_y), f_{\bar{y}}(h(\boldsymbol{x}_1), \boldsymbol{z}))), s_y(h(\boldsymbol{x}_y), \boldsymbol{x}_y)) \right|_1 \right],$$

$$\mathcal{L}_{\text{cyc}}^s := \mathbb{E}_y \left[ \mathbb{E}_{\boldsymbol{x}_y} \mathbb{E}_{\boldsymbol{x}_{\bar{y}} \sim \mathbb{P}_{x_{\bar{y}}|h(x_y)}} \left| \boldsymbol{x}_y - g(e_{\bar{y}}(g(e_y(\boldsymbol{x}_y), s_{\bar{y}}(h(\boldsymbol{x}_1), \boldsymbol{x}_{\bar{y}}))), s_y(h(\boldsymbol{x}_y), \boldsymbol{x}_y)) \right|_1 \right]. \tag{5}$$

---

[2]Different of $d$, the embedding function, that we introduced in the previous subsection.

**Style-consistency loss** (Almahairi et al., 2018; Huang et al., 2018). Regularizes the translation networks to use the style code.

$$
\begin{aligned}
\mathcal{L}_{\mathrm{sty}}^f &:= \mathbb{E}_y\left[\mathbb{E}_{\boldsymbol{x}_y}\mathbb{E}_{\boldsymbol{z}}\left|f_{\bar{y}}(h(\boldsymbol{x}_y),\ \boldsymbol{z}) - s_{\bar{y}}(h(\boldsymbol{x}_y),\ g(e_y(\boldsymbol{x}_y),\ f_{\bar{y}}(h(\boldsymbol{x}_y),\ \boldsymbol{z})))\right|_1\right], \\
\mathcal{L}_{\mathrm{sty}}^s &:= \mathbb{E}_y\left[\mathbb{E}_{\boldsymbol{x}_y}\mathbb{E}_{\boldsymbol{x}_{\bar{y}}\sim\mathbb{P}_{x_{\bar{y}}|h(\boldsymbol{x}_y)}}\left|s_{\bar{y}}(h(\boldsymbol{x}_y),\ x_{\bar{y}}) - s_{\bar{y}}(h(\boldsymbol{x}_y),\ g(e_y(\boldsymbol{x}_y),\ s_{\bar{y}}(h(\boldsymbol{x}_y),\ x_{\bar{y}})))\right|_1\right].
\end{aligned}
\tag{6}
$$

**Style diversity loss** (Yang et al., 2019; Choi et al., 2017). Regularizes the translation network to produce diverse samples.

$$
\begin{aligned}
\mathcal{L}_{\mathrm{sd}}^f &:= -\mathbb{E}_y\left[\mathbb{E}_{x_y}\mathbb{E}_{\boldsymbol{z},\boldsymbol{z}'}\left|g(e_y(\boldsymbol{x}_y),\ f_{\bar{y}}(h(\boldsymbol{x}_y),\ \boldsymbol{z})) - g(e_y(\boldsymbol{x}_y),\ f_{\bar{y}}(h(\boldsymbol{x}_y),\ \boldsymbol{z}'))\right|_1\right], \\
\mathcal{L}_{\mathrm{sd}}^s &:= -\mathbb{E}_y[\mathbb{E}_{x_y}\mathbb{E}_{\boldsymbol{x}_{\bar{y}},\boldsymbol{x}_{\bar{y}}'\sim\mathbb{P}_{x_{\bar{y}}|h(x_y)}}\left|g(e_y(\boldsymbol{x}_y,\ s_{\bar{y}}(h(\boldsymbol{x}_y),\boldsymbol{x}_{\bar{y}}))) - g(e_y(\boldsymbol{x}_y,s_{\bar{y}}(h(\boldsymbol{x}_y),\boldsymbol{x}_{\bar{y}}')))\right|_1]
\end{aligned}
\tag{7}
$$

**Semantic loss.** We introduce the following semantic loss as the cross-entropy between the semantic code of the source samples and that of their corresponding generated samples. We use this loss to regularise the generation to be semantically coherent with the source input.

$$
\begin{aligned}
\mathcal{L}_{\mathrm{sem}}^f &:= -\mathbb{E}_y\left[\mathbb{E}_{\boldsymbol{x}_y,\boldsymbol{z}}[h(\boldsymbol{x}_y)\log(h(g(e_y(\boldsymbol{x}_y),\ f_{\bar{y}}(h(\boldsymbol{x}_y),\ \boldsymbol{z}))))]\right], \\
\mathcal{L}_{\mathrm{sem}}^s &:= -\mathbb{E}_y\left[\mathbb{E}_{\boldsymbol{x}_y}\mathbb{E}_{\boldsymbol{x}_{\bar{y}}\sim\mathbb{P}_{x_{\bar{y}}|h(x_y)}}[h(\boldsymbol{x}_y)\log(h(g(e_y(\boldsymbol{x}_y),\ s_{\bar{y}}(h(\boldsymbol{x_y}),\ \boldsymbol{x}_{\bar{y}}))))]\right].
\end{aligned}
\tag{8}
$$

Finally, we combine all our losses and solve the following optimization.

$$
\begin{aligned}
\arg\min_{g,e.,f.,s.}\arg\max_{d.}\ &\mathcal{L}_{\mathrm{adv}}^s + \mathcal{L}_{\mathrm{adv}}^f + \lambda_{\mathrm{sty}}(\mathcal{L}_{\mathrm{sty}}^s + \mathcal{L}_{\mathrm{sty}}^f) + \lambda_{\mathrm{cyc}}(\mathcal{L}_{\mathrm{cyc}}^s + \mathcal{L}_{\mathrm{cyc}}^f) + \\
&\lambda_{\mathrm{sd}}(\mathcal{L}_{\mathrm{sd}}^s + \mathcal{L}_{\mathrm{sd}}^f) + \lambda_{\mathrm{sem}}(\mathcal{L}_{\mathrm{sem}}^s + \mathcal{L}_{\mathrm{sem}}^f),
\end{aligned}
\tag{9}
$$

where $\lambda_{\mathrm{sty}} > 0$, $\lambda_{\mathrm{cyc}} > 0$, $\lambda_{\mathrm{sd}} > 0$ and $\lambda_{\mathrm{sem}} > 0$ are hyper-parameters defined as the weight of each losses.

## 4 EXPERIMENTS

We compare CatS-UDT with other *unsupervised* domain translation methods and demonstrate that it shows significant improvements on the SPUDT and SHDT problems. We then perform ablation and comparative studies to investigate the cause of the improvements on both setups. We demonstrate SPUDT using the MNIST (LeCun & Cortes, 2010) and SVHN (Netzer et al., 2011) datasets and SHDT using Sketches and Reals samples from the DomainNet dataset (Peng et al., 2019). We present the datasets in more detail and the baselines in Appendix B.1 and Appendix B.2 respectively.

### 4.1 SPUDT WITH MNIST↔SVHN

**Adapted clustering.** We first cluster MNIST using IMSAT (Hu et al., 2017). We reproduce the accuracy of 98.24%. Using the learned clusters as ground-truth labels for MNIST, we adapt the clusters using the VMT (Mao et al., 2019) framework for unsupervised domain adaptation. This trained classifier achieves an accuracy of 98.20% on MNIST and 88.0% on SVHN. See Appendix B.3 and Appendix B.5 for more details on the methods used.

**Evaluation.** We consider two evaluation metrics for SPUDT. (1) *Domain translation accuracy*, to indicate the proportion of generated samples that have the same semantic category as the source

Table 1: **Comparison with the baselines.** Domain translation accuracy and FID obtained on MNIST (M) ↔SVHN (S) for the different methods considered. The last column is the test classification accuracy of the classifier used to compute the metric. *: Using weak supervision.

|  | Data | CycleGAN | MUNIT | DRIT | Stargan-V2 | EGSC-IT* | CatS-UDT | Target |
|---|---|---|---|---|---|---|---|---|
| Acc | M→S | 10.89 | 10.44 | 13.11 | 28.26 | 47.72 | **95.63** | 98.0 |
| | S→M | 11.27 | 10.12 | 9.54 | 11.58 | 16.92 | **76.49** | 99.6 |
| FID | M→S | 46.3 | 55.15 | 127.87 | 66.54 | 72.43 | **39.72** | - |
| | S→M | 24.8 | 30.34 | 20.98 | 26.27 | 19.45 | **6.60** | - |

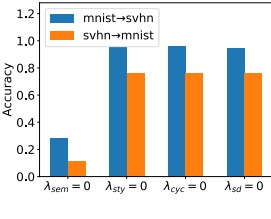 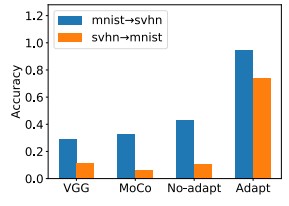 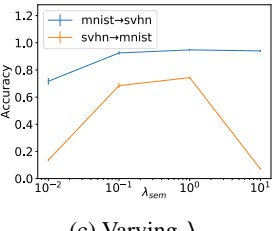

(a) Ablation study.  (b) Training of semantic encoder.  (c) Varying $\lambda_{\text{sem}}$.

Figure 3: **Studies** on the effect on the translation accuracy on MNIST↔SVHN of (a) Ablating each loss by setting their $\lambda = 0$. (b) Using VGG, MoCO, the presented method for learning categorical semantics without adaptation and with adaptation respectively to train a semantic encoder. (c) Varying $\lambda_{\text{sem}}$.

samples. To compute this metric, we first trained classifiers on the target domains. The classifiers obtain an accuracy of 99.6% and 98.0% on the test set of MNIST and SVHN respectively – as reported in the last column of Table 1. (2) FID (Heusel et al., 2017) to evaluate the generation quality.

**Comparison with the baselines.** In Table 1, we show the test accuracies obtained on the baselines as well as with CatS-UDT. We find that all of the UDT baselines perform poorly, demonstrating the issue of translating samples through a large domain-shift without supervision. However, we do note that StarGAN-V2 obtains slightly higher than chance numbers for MNIST→SVHN. We attribute this to a stronger implicit bias toward identity. EGSC-IT, which uses supervised labels, shows better than chance results on both MNIST→SVHN and SVHN→MNIST, but not better than CatS-UDT.

**Ablation study – the effect of the losses** In Figure 3a, we evaluate the effect of removing each of the losses, by setting their $\lambda = 0$, on the translation accuracy. We observe that the semantic loss provides the biggest improvement. We run the same analysis for the FID in Appendix C.2 and find the same trend. The integration of the semantic loss, therefore, improves the preservation of semantics in domain translation and also improves the generation quality. We also inspect more closely $\lambda_{\text{sem}}$ and evaluate the effect of varying it in Figure 3c. We observe a point of diminishing returns, especially for SVHN→MNIST. We observe that the reason for this diminishing return is that for a $\lambda_{\text{sem}}$ that is too high, the generated samples resemble a mixture of the source and the target domains, rendering the samples out of the distribution in comparison to the samples used to train the classifier used for the evaluation. We demonstrate this effect and discuss it in more detail in Appendix C.2 and show the same diminishing returns for the FID.

**Comparative study – the effect of the semantic encoder.** In Figure 3b, we evaluate the effect of using a semantic encoder trained using a VGG (Liu & Deng, 2015) on classification, using a ResNet50 on MoCo (He et al., 2020), to cluster MNIST but not adapted to SVHN and to cluster MNIST with adaptation to SVHN. We observe that the use of an adapted semantic network improves the accuracy over its non-adapted counterpart. In Appendix C.2 we present the same plot for the FID. We also observe that the FID degrades when using a non-adapted semantic encoder. Overall, this demonstrates the importance of adapting the network inferring the semantics, especially when the domains are sufficiently different.

CycleGAN   DRIT   EGSC-IT  StarGAN-v2  CatS-UDT (ours)

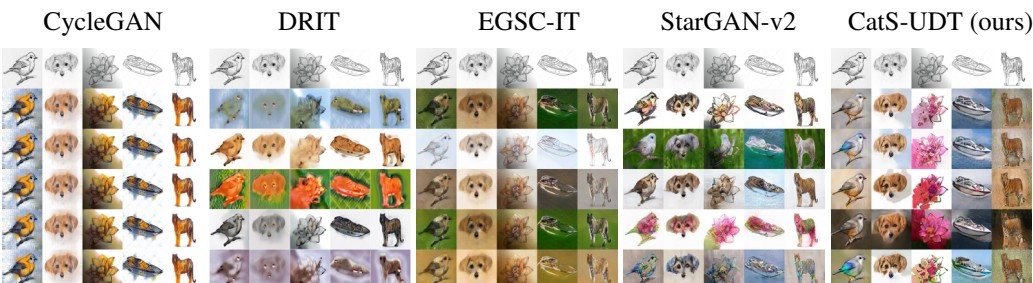

Figure 4: **Comparison with baselines.** Comparing the baselines with our approach for translating sketches to real images. For each sketch (top row), we sample 5 different styles generating 5 images in the target domain. For CycleGAN, we copy the generated images 5 times because it is impossible to generate multiple samples in the target domain from the same source image.

Table 2: **Comparison with the baselines.** Comparing the FID obtained on Sketch→Real for the baselines and our method. We compute the FID per class and over all the categories.

| DATA | CYCLEGAN | DRIT | EGSC-IT | STARGAN-V2 | CATS-UDT (OURS) |
|------|----------|------|---------|------------|-----------------|
| BIRD | 124.10 | 141.18 | 101.09 | 93.58 | **92.69** |
| DOG | 170.12 | 153.05 | 145.18 | 108.62 | **105.59** |
| FLOWER | 242.84 | 223.63 | 225.24 | 209.91 | **137.01** |
| SPEEDBOAT | 189.20 | 239.94 | 174.78 | 127.23 | **126.18** |
| TIGER | 156.54 | 245.73 | 109.97 | 69.08 | **41.77** |
| ALL | 102.37 | 128.45 | 86.86 | 65.00 | **58.69** |

## 4.2 SHDT WITH SKETCHES→REALS

**Adapted clustering.** The representations of the real images were obtained by using MoCo-V2 – a self-supervised model – pre-trained on unlabelled ImageNet. We clustered the learned representation using spectral clustering (Donath & Hoffman, 1973; Luxburg, 2007), yielding 92.13% clustering accuracy on our test set of real images. Using the learned cluster as labels for the real images, we adapted our clustering to the sketches by using a domain adaptation framework – VMT (Mao et al., 2019) – on the representation of the sketches and the reals. This process yields an accuracy of 75.47% on the test set of sketches and 90.32% on the test set of real images. More details are presented in Appendix B.4 and in Appendix B.5.

**Evaluation.** For the Sketch→Real experiments, we evaluate the quality of the generations by computing the FID over each class individually as well as over all the classes. We do the former because the translation network may generate realistic images that are semantically unrelated to the sketch translated.

**Comparison with baselines.** We depict the issue with the UDT baselines in Figure 4. For DRIT and StarGAN-V2, the style is independent of the source image. CycleGAN does not have this issue because it does not sample a style. However, the samples are not visually appealing. The images generated with EGSC-IT are dependent on the source, but the style is not realistic for all the source categories. We quantify the difference in sample quality in Table 2 where we present the FIDs.

**Ablation study – the effect of the losses.** In Table 3a, we evaluate the effect of setting each of removing each of the losses, by setting their $\lambda = 0$, on the FIDs on Sketches→Reals. As in SPUDT, the semantic loss plays an important role. In this case, the semantic loss encourages the network to use the semantic information. This can be visualized in Appendix C.3 where we plot the translation. We see that $\lambda_{sem} = 0$ suffers from the same problem that the baselines suffered, that is that the style is not relevant to the semantic of the source sample.

**Comparative study – the effect of the methods to condition semantics.** We compare different methods of using semantic information in a translation network, in Table 3b. *None* refers to the case where the semantics is not explicitly used in the translation network, but a semantic loss is still used. This method is commonly used in supervised domain translation methods such as Bousmalis et al.

Table 3: **Studies** on the effect of the translation accuracy on Sketches→Reals on (a) Ablating each loss by setting their coefficient $\lambda = 0$. (b) Methods to condition the translation network on the semantics: Not conditioning, conditioning the content representation with categorical semantics, conditioning the content representation with VGG, and conditioning the style with categorical semantics.

(a) Ablation study of the losses

| Data | $\lambda_{sem} = 0$ | $\lambda_{sty} = 0$ | $\lambda_{cyc} = 0$ | $\lambda_{SD} = 0$ |
|------|------|------|------|------|
| Bird | 148.32 | 94.18 | 108.68 | 101.97 |
| Dog | 131.35 | 109.50 | 120.39 | 106.24 |
| Flower | 211.84 | 124.37 | 160.97 | 154.77 |
| Speedboat | 185.11 | 97.52 | 127.68 | 99.67 |
| Tiger | 153.03 | 39.24 | 52.64 | 41.55 |
| All | 69.19 | 53.43 | 67.88 | 58.47 |

(b) Method to condition on the semantics.

| Data | None | Content | Content(VGG) | Style |
|------|------|---------|--------------|-------|
| Bird | 101.88 | 405.29 | 129.69 | 92.69 |
| Dog | 142.79 | 343.62 | 229.18 | 105.59 |
| Flower | 196.70 | 323.52 | 220.72 | 137.01 |
| Speedboat | 160.57 | 280.47 | 192.38 | 126.18 |
| Tiger | 57.29 | 212.69 | 228.84 | 41.77 |
| All | 81.69 | 275.21 | 112.10 | 58.59 |

(2017); Hoffman et al. (2018); Tomei et al. (2019). *Content* refers to the case where we use categorical semantics, inferred using our method, to condition the content representation. Similarly, we also consider the method used in Ma et al. (2019), in which the semantics comes from a VGG encoder trained with classification. We label this method *Content(VGG)*. For these two methods, we learn a mapping from the semantic representation vector to a feature-map of the same shape as the content representation and then multiply them element-wise – as done in EGSC-IT. *Style* refers the presented method to modulate the style. First, for *None*, the network generates only one style per semantic class. We believe that the reason is that the semantic loss penalizes the network for generating samples that are outside of the semantic class, but the translation network is agnostic of the semantic of the source sample. Second, for *Content*, the network fails to generate sensible samples. The samples are reminiscent of what happens when the content representation is of small spatial dimensionality. This failure does not happen for *Content(VGG)*. Therefore, from the empirical results, we conjecture that the failure case is due to a large discrepancy between the content representation and the categorical representation in addition to a pressure from the semantic loss. The semantic loss forces the network to use the semantic incorporated in the content representation, thereby breaking the spatial structure. This demonstrates that our method allows us to incorporate the semantics category of the source sample without affecting the inductive bias toward the identity, in this setup.

## 5 CONCLUSION AND DISCUSSION

We discussed two situations where the current methods for UDT are found to be lacking - Semantic Preserving Unsupervised Domain Translation and Style Heterogeneous Domain Translation. To tackle these issues, we presented a method for learning domain invariant categorical semantics without supervision. We demonstrated that incorporating domain invariant categorical semantics greatly improves the performance of UDT in these two situations. We also proposed to condition the style on the semantics of the source sample and showed that this method is beneficial for generating a style related to the semantic category of the source sample in SHDT, as demonstrated in Sketches→Reals.

While we demonstrated that using domain invariant categorical semantics improves the translation in the SPUDT and SHDT settings, we re-iterate that the quality of the network used to infer the semantics is important. We observe an example of the effect of mis-clustering the source sample in Figure 12e third column: the tiger is incorrectly translated into a dog. This failure may potentially translate to any application, even outside UDT, using learned categorical semantics. Efforts on robust machine learning and detections of failures are also important in this setup for countering this failure.

## ACKNOWLEDGMENTS

We would like to acknowledge Devon Hjelm, Sébastien Lachapelle, Jacob Leygoni and Amjad Almahairi for the helpful discussions. We also acknowledge Compute-Canada and Mila for providing the computing ressources used for this work. This work would not have been possible without the development of the open-source softwares that we used. This includes: Python, Pytorch (Paszke et al., 2019), Numpy (Harris et al., 2020) and Matplotlib (Hunter, 2007). We acknowledge financial support of Hitachi, Samsung, CIFAR and the Natural Sciences and Engineering Research Council of Canada (NSERC Discovery Grant).

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

## A  BACKGROUND CROSS-DOMAIN SEMANTICS LEARNING

**Unsupervised representation learning** aims at learning an embedding of the input which will be useful for a downstream task, without direct supervision about the task(s) of interest. For a downstream task of classification, success is typically defined, but not limited, as the ability to classify the learned representation with a linear classifier. Recent advances have produced very impressive results by exploiting self-supervision, where a useful supervisory signal is concocted from within the unlabeled dataset. Contrastive learning methods such as CPC (van den Oord et al., 2018a), DIM (Hjelm et al., 2019), SimCLR (Chen et al., 2020a), and MoCo (He et al., 2020; Chen et al., 2020b) have shown very strong success, for example achieving more than 70% top-1 accuracy on ImageNet by linear classification on the learned embeddings.

**Clustering** separates data (or a representation of the data) into an $N$-discrete set. $N$ can be known *a priori*, or not. While methods such as K-means (Lloyd, 2006) and spectral clustering (Luxburg, 2007) are classic, recent deep learning approaches such as DEC (Xie et al., 2016), IMSAT (Hu et al., 2017) and Deep Clustering (Caron et al., 2018) demonstrate that the representation of a neural network can be used for clustering complex, high-dimensional data.

**Unsupervised domain adaptation** (Ben-David et al., 2010) aims at adapting a classifier from a labelled source domain to an unlabelled target domain. Ganin et al. (2016) uses the gradient reversal method to minimize the divergence between the hidden representations of the source and target domains. Follow-up methods have been proposed to adapt the classifier by relevant regularization; VADA (Shu et al., 2018) regularizes using the cluster assumption (Grandvalet & Bengio, 2005) and virtual adversarial training (Miyato et al., 2018), VMT (Mao et al., 2019) suggests using virtual Mixup training.

## B  ADDITIONAL EXPERIMENTAL DETAILS

Our results on MNIST↔SVHN and Sketches→Reals datasets were obtained using our Pytorch (Paszke et al., 2019) implementation. We provide the code which contains all the details necessary for reproducing the results as well as scripts that will themselves reproduce the results.

Here, we provide additional experimental and technical details on the methods used. In particular, we present the datasets and the baselines used. We follow with a detailed background on IMSAT (Hu et al., 2017) which is used to learn a clustering on MNIST in our MNIST↔SVHN. Next, we give a background on MoCO (Chen et al., 2020b) which is used to learn a representation on the Reals. Then, we provide a background on Virtual Mixup Training, which is the domain adaptation technique that we use to adapt either the MNIST to SVHN or Reals to Sketches. Finally, we provide a method for evaluating the clusters across multiple domains.

### B.1  EXPERIMENTAL DATASETS

Throughout our SPUDT experiments, we transfer between both the **MNIST** (LeCun & Cortes, 2010), which we upsample to $32 \times 32$ and triple the number of channels, and the **SVHN** (Netzer et al., 2011) datasets. We don't alter the SVHN dataset, i.e. we consider $32 \times 32$ samples with 3 channels RGB without any data augmentation. But, we consider samples with feature values in the range [-1, 1], as it is usually done in the GAN litterature (Radford et al., 2015), for all of our datasets.

We use a subset of **Sketches** and **Reals** from the DomainNet dataset (Peng et al., 2019) to demonstrate the task of SHDT. We use the following five categories of the DomainNet dataset: *bird, dog, flower, speedboat and tiger*; these 5 are among the categories with most samples in both our domains and possessing distinct styles which are largely non-interchangeable. We resized every image to $256 \times 256$.

### B.2  BASELINES

For our UDT baselines, we compare with CycleGAN (Zhu et al., 2017b), MUNIT (Huang et al., 2018), DRIT (Lee et al., 2019) and StarGAN-V2 (Choi et al., 2019). We use these baselines because they are, to our knowledge, the reference models for unsupervised domain translation today. But, none of these baselines use semantics. Also, we are not aware of any UDT method that proposes to

use semantics without supervision. Hence, we also consider EGST-IT (Ma et al., 2019) as a baseline although it is weakly supervised by the usage of a pre-trained VGG network. EGSC-It proposes to include the semantics into the translation network by conditioning the content representation. It also considers the usage of exemplar, unconditionally of the source sample.

For each of the baselines, we perform our due diligence to find the set of parameters that perform the best and report our results using these parameters.

### B.3 IMSAT FOR CLUSTERING MNIST

**Clustering**  Following RIM (Gomes et al., 2010) and IMSAT (Hu et al., 2017), we learn a mapping $c : \mathcal{X} \to \mathcal{C}$, where $\mathcal{C} \in \mathbb{R}^k$ is a continuous space representing a soft clustering of $\mathcal{X}$, by optimizing the following objective

$$\min_c \lambda \mathcal{R}(c) - I(\mathcal{X}; \mathcal{C}), \tag{10}$$

where $\lambda > 0$ is a Lagrange multiplier, $I$ is the mutual information defined as

$$I(\mathcal{X}; \mathcal{C}) = H(\mathcal{C}) - H(\mathcal{C}|\mathcal{X}),$$

and $\mathcal{R}$ is a regularizer to restrict the class of functions. As in IMSAT, we use the regularizer

$$\mathcal{R}(c) = \mathbb{E}_{\boldsymbol{x} \sim \mathbb{P}_x} ||c(\boldsymbol{x}) - c(\boldsymbol{x}')||_2^2, \tag{11}$$

where $\boldsymbol{x}' = T(\boldsymbol{x})$, and $T$ is a set of transformations of the original image, such as affine transformations. Essentially, this ensures that the mapping is invariant under the set of transformations defined by $T$. In particular, we used affine translations such as rotation scaling and skewing.

If $c$ is a deterministic function, then $H(\mathcal{C}|\mathcal{X}) = 0$, and $\max I(\mathcal{X}; \mathcal{C}) = \max H(\mathcal{C})$. Hence, we are interested in a clustering of maximum entropy. This can be achieved if $\mathbb{P}_c = \mathbb{P}_{\text{cat}}$ where $\mathbb{P}_{\text{cat}}$ is the categorical distribution with uniform probability for every category (or a prior distribution, if we have access to it).

Thus, we can maximize the mutual information by mapping to the uniform categorical distribution. IMSAT minimizes the KL-divergence, $D_{\text{KL}}[\mathbb{P}_c || \mathbb{P}_{\text{cat}}]$. Equivalently, we can minimize the EMD $W_1(c\#\mathbb{P}_x, \mathbb{P}_{\text{cat}})$ using the Wasserstein GAN framework, where $\#$ denotes the *push-forward* function.

$$I(\mathcal{X}; \mathcal{C}) = \max_{f:\text{Lipschitz-1}} E_{\hat{\boldsymbol{u}} \sim c\#\mathbb{P}_x} f(\hat{\boldsymbol{u}}) - \mathbb{E}_{\boldsymbol{u} \sim \mathbb{P}_{\text{cat}}} f(\boldsymbol{u}). \tag{12}$$

Using equation 11 and equation 12 in equation 10, we obtain the following objective for clustering

$$\min_c \max_{f:\text{Lipschitz-1}} \lambda \mathbb{E}_{\boldsymbol{x} \sim \mathbb{P}_x} ||c(\boldsymbol{x}) - c(\boldsymbol{x}')||_2 - \mathbb{E}_{\hat{\boldsymbol{u}} \sim c\#\mathbb{P}_x} f(\hat{\boldsymbol{u}}) - \mathbb{E}_{\boldsymbol{u} \sim \mathbb{P}_{\text{cat}}} f(\boldsymbol{u}).$$

### B.4 SELF-SUPERVISION OF REAL IMAGES WITH MoCo

We use MoCo (He et al., 2020), a self-supervised representation-learning algorithm, for learning an embedding from the sketches and reals images to a code.

Let $f_q$ and $f_k$ be two networks. Assume that $f_k$ is a moving average of $f_q$. MoCo principally minimizes the following contrastive loss, called InfoNCE (van den Oord et al., 2018b), with respect to the parameters of $f_q$.

$$\mathcal{L}_{\text{nce}} = -\log \frac{\exp(f_q(\boldsymbol{x}_i) \cdot f_k(\boldsymbol{x}_i)/\tau)}{\sum_{j=0}^{K} \exp(f_q(\boldsymbol{x}_i) \cdot f_k(\boldsymbol{x}_j)/\tau)}. \tag{13}$$

The parameters $\theta_k$ of $f_k$ are updates as follows

$$\theta_k \leftarrow m\theta_k + (1 - m)\theta_q$$

where $m \in [0, 1)$ and $\theta_q$ are the parameters obtained by minimizing equation 13 by gradient descent.

Furthermore, a dictionary of the representation $f_k(\boldsymbol{x})$ is preserved and updated throughout the training, allowing to have more *negative samples*, i.e., $K$ in equation 13 can be bigger. We refer to the main paper for more technical details.

### B.5 VIRTUAL MIXUP TRAINING FOR UNSUPERVISED DOMAIN ADAPTATION

Domain adaptation aims at adapting a function trained on a domain $\mathcal{X}$ so that it can perform well on a domain $\mathcal{Y}$. Unsupervised domain adaptation refers to the case where the target domain $\mathcal{Y}$ is unlabelled during training. Normally, it assumes supervised labels on the source domain. Here, we will instead assume that we have a pre-trained inference network $c$ trained to cluster $\mathcal{X}$. In other words, we do not assume ground truth labels. For MNIST and SVHN, we consider $\mathcal{X}$ and $\mathcal{Y}$ to be the raw images. For Sketches and Reals, we consider $\mathcal{X}$ and $\mathcal{Y}$ to be their learned embeddings.

It has been shown that the error of a hypothesis function $h$ on the target domain $\mathcal{Y}$ is upper bounded by the following (Ben-David et al., 2010)

$$\mathcal{L}_{\mathcal{Y}}(h) \leq \mathcal{L}_{\mathcal{X}}(h) + d(\mathcal{X}, \mathcal{Y}) + \min_{h'} \mathcal{L}_{\mathcal{X}}(h') + \mathcal{L}_{\mathcal{Y}}(h'),$$

where $\mathcal{L}_x$ is the risk and can be computed given a loss function, for example the cross entropy. Then

$$\mathcal{L}_{\mathcal{X}}(h) = -\mathbb{E}_{\boldsymbol{x} \sim \mathbb{P}_x} c(\boldsymbol{x}) \log h(\boldsymbol{x}).$$

Lately, unsupervised domain adaptation has seen major improvements. In this work, we shall leverage the tricks proposed in Ganin et al. (2016); Shu et al. (2018); Mao et al. (2019) because of their demonstrated empirical success in the modalities that interest us in this work. We briefly describe these techniques below.

**Gradient reversal**  Initially proposed in Ganin et al. (2016), gradient reversal aims to match the marginal distribution of intermediate hidden representations of a neural network across domains. If $h$ is a neural network and can be composed as $h = h_2 \circ h_1$, then gradient reversal is defined as

$$\mathcal{L}_{\text{gv}}(h_1) = \max_{D} \ \mathbb{E}_{\boldsymbol{h}_x \sim h_1 \# \mathbb{P}_x}[\log D(\boldsymbol{h}_x)] + \mathbb{E}_{\boldsymbol{h}_y \sim h_1 \# \mathbb{P}_y}[\log(1 - D(\boldsymbol{h}_y))].$$

which can also be seen as applying a GAN loss (Goodfellow et al., 2014) on a representation of a neural network.

**Cluster assumption**  The cluster assumption (Chapelle & Zien, 2005) is simply an assumption that the data is clusterable into classes. In other words, it states that the decision boundaries of $h$ should be in low-density regions of the data. To encourage this, Grandvalet & Bengio (2005) propose to minimize the following objective on the conditional entropy:

$$\mathcal{L}_c(h) = -\mathbb{E}_{\boldsymbol{y} \sim \mathbb{P}_y} h(\boldsymbol{y}) \log h(\boldsymbol{y}).$$

However, in practice, such a constraint is applied on an empirical distribution. Hence, nothing stops the classifier from abruptly changing its predictions for any samples outside of the training distribution. This motivates the next constraints.

**Virtual adversarial training**  Shu et al. (2018) propose to alleviate this problem by constraining $h$ to be locally-Llipschitz around an $\epsilon$-ball. Borrowing from Miyato et al. (2018), they propose the additional regularizer

$$\mathcal{L}_{vx}(h) = \max_{||\boldsymbol{r}||_2 \leq \epsilon} D_{\text{KL}}(h(\boldsymbol{x}) \,||\, h(\boldsymbol{x} + \boldsymbol{r})),$$

with $\epsilon > 0$.

**Virtual mixup training**  With similar motivations, Mao et al. (2019) propose that the prediction of an interpolated point $\tilde{\boldsymbol{x}}$ should itself be an interpolation of the predictions at $\boldsymbol{x}_1$ and at $\boldsymbol{x}_2$. We compute interpolates as

$$\tilde{\boldsymbol{x}} = \alpha \boldsymbol{x}_1 + (1 - \alpha)\boldsymbol{x}_2,$$
$$\tilde{\boldsymbol{y}} = \alpha h(\boldsymbol{x}_1) + (1 - \alpha)h(\boldsymbol{x}_2),$$

with $\alpha \sim U(0, 1)$, where $U(0, 1)$ is a continuous uniform distribution between 0 and 1.

The proposed objective is then simply

$$\mathcal{L}_{mx}(h) = -\mathbb{E}_{\boldsymbol{x} \sim \mathbb{P}_x} \tilde{\boldsymbol{y}}^{\top} \log h(\tilde{\boldsymbol{x}}).$$

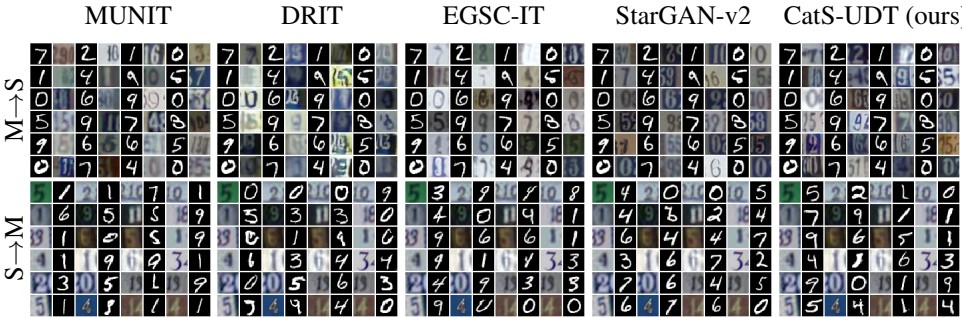

Figure 5: Qualitative comparison of the baselines with our method on MNIST↔SVHN. Even columns correspond to source samples, and odd columns correspond to their translations.

These objectives are composed to give the overall optimization problem:

$$\min_h \mathcal{L}_{\mathcal{X}}(h) + \lambda_1 \mathcal{L}_{\text{gv}}(h_1) + \lambda_2 \mathcal{L}_{\text{c}}(h) + \lambda_3 \mathcal{L}_{\text{vx}}(h) + \lambda_4 \mathcal{L}_{\text{vy}}(h) + \lambda_5 \mathcal{L}_{\text{mx}}(h) + \lambda_6 \mathcal{L}_{\text{my}}(h),$$

where the second subscript $x$ or $y$ denotes the domain.

Finally, we note that for MNIST↔SVHN, we perform the adaptation directly on the image space. For Sketches→Reals, we found that it worked better to perform the adaptation on the representation space instead.

### B.6 Evaluation of the learned cross-domain clusters

An important detail is the evaluation of the clustering across the domains. This evaluation indeed gives a signal of how good the categorical representation is. But, because the cluster identities might have been shifted from the pre-defined labels in the validation set, it is important to consider this shift when performing the evaluation. Therefore, we first define the correspondence for the cluster to label $\tilde{l}_k$ as follows

$$\tilde{l}_k = \underset{j \in \{0,\ldots,|\mathcal{C}|-1\}}{\arg\max} \sum_{i=1}^{N} \mathbb{I}(j, l_i)|k \cap c(\boldsymbol{x}_i)_j|. \tag{14}$$

where $\mathbb{I}$ is the indicator function returning 1 if $j = l_i$ and 0 otherwise. Essentially, equation 14 is necessary because we want the same labels in both domains to map to the same cluster. Hence, simply computing the purity evaluation could be misleading in the case where both domains are clustered correctly, but the clusters do not align to the same labels. Using this correspondence, we can now proceed to evaluate the clustering adaptation using the evaluation accuracy as one would normally do.

## C Additional results

### C.1 Qualitative results for MNIST-SVHN

We present additional qualitative results to provide a better sense of the results that our method achieves. In Figure 5, we show qualitative comparisons with samples of translation for the baselines and our technique. We observe that the use of semantics in the translation visibly helps with preserving the semantic of the source samples. The qualitative results confirm the quantitative results on the preservation of the digit identity presented in Table 1.

Furthermore, in Figure 6, we present qualitative results of the effect of changing the noise sample $z$ on the generation of SVHN samples for the same MNIST source sample. The first row represents the source samples and each column represents a generation with a different $z$. Each source sample uses the same set of $z$ in the same order. We observe that $z$ indeed grossly controls the style of the generation. Also, we observe that the generations preserve features of the source sample such as the pose. However, we note that some attributes such as typography are not perfectly preserved. In this instance, we conjecture that this is due to the fact the the "MNIST typography" is not the same as the "SVHN typography". For example, the '4's are different in the MNIST and SVHN datasets. Therefore, due to the adversarial loss, the translation has to modify the typography of MNIST.

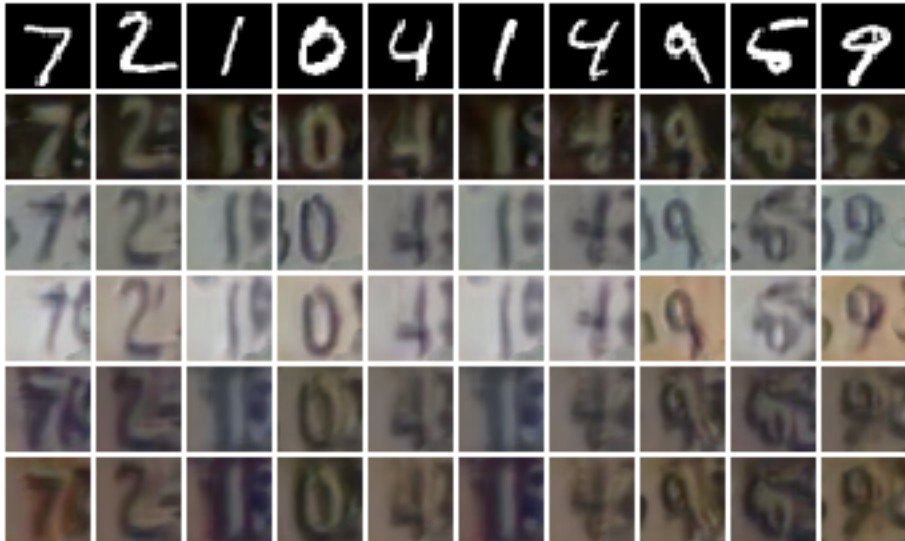

Figure 6: Multiple sampling for MNIST→SVHN. For each column, the first row is the source sample and each subsequent row is a generation corresponding to a different $z$.

### C.2 ADDITIONAL ABLATION STUDIES FOR MNIST-SVHN

**Ablation study – the effect of the losses on the FID.** In Figure 7a, we evaluate the effect of removing each of the losses, by setting their $\lambda = 0$, on the FID. We observe that removing the semantic loss yields the biggest deterioration for the FID. Hence, the semantic loss does not only improve the semantic preservation as observed in Section 4.1, but also the image quality of the translation.

Also, we see a U-curve on the FID on MNIST→SVHN with respect to the parameter $\lambda_{\text{sem}}$. We observe that tuning this parameter allows us to improve the generation quality. We make a similar observation for SVHN→MNIST for both the FID and the accuracy. In Figure 7c, we present qualitative results of the effect of setting $\lambda_{\text{sem}} = 10$. We see that the samples are a mix of MNIST and SVHN samples. The reduction in generation quality explains why we obtain a worst FID when $\lambda_{\text{sem}}$ is too high. Moreover, we see that the generated samples are out-of-distribution, explaining why we obtain a low accuracy although the digit identity is preserved.

**Comparative study – effect of the method to condition the semantics.** In Figure 8a and in Figure 8b, we evaluate the effect of the method to condition the semantics – in MNIST↔SVHN – on the translation accuracy and on the FID respectively.

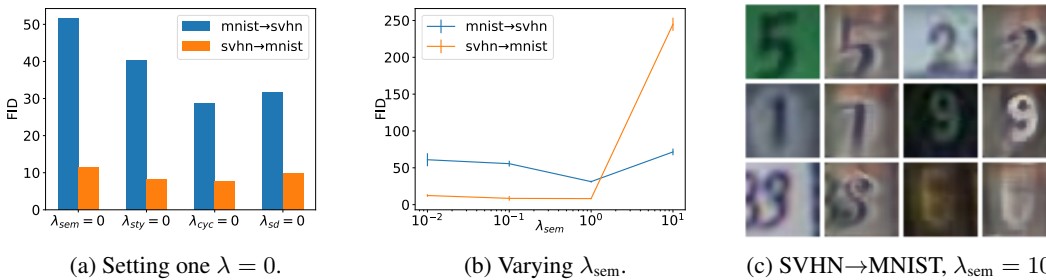

(a) Setting one $\lambda = 0$.     (b) Varying $\lambda_{\text{sem}}$.     (c) SVHN→MNIST, $\lambda_{\text{sem}} = 10$

Figure 7: Ablation studies on the effect on the FID on MNIST↔SVHN of (a) Setting one $\lambda = 0$ while keeping the other $\lambda' = 1$, (b) Varying $\lambda_{\text{sem}}$ and (c) Qualitative results of SVHN→ MNIST when $\lambda_{\text{sem}} = 10$.

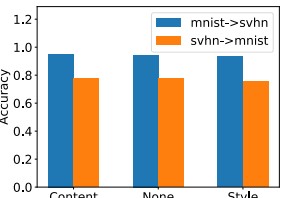 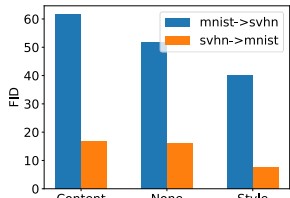 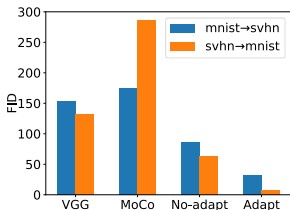

(a) **Accuracy** of conditioning methods.  (b) **FID** of conditioning methods.  (c) **FID** of semantic encoders.

Figure 8: Comparative studies on the effect (a) on the translation accuracy and (b, c) on the FID on MNIST↔SVHN on (a, b) Conditioning the content representation on the semantics, not conditioning on semantics and conditioning the style representation on the semantics.

*None* refers to the case where the semantics is not explicitly used to condition any part of the translation network, but the semantic loss is still used. This method is commonly used in supervised domain translation methods such as Bousmalis et al. (2017); Hoffman et al. (2018); Tomei et al. (2019). *Content* refers to the case where categorical semantics are used to condition the content representation. This method is similar to the method used in Ma et al. (2019), for example, with the exception that the semantic encoder they used is a VGG trained on a classification task. *Style* refers to the case where the categorical semantics are used to condition the style, as we propose to do.

We see that the method to condition the semantics does not affect the translation accuracy on MNIST↔SVHN. However, it does affect the generation quality. This further demonstrates the relevance of injecting the categorical semantics by modulating the style of the generated samples.

**Comparative study – effect of adapting the categorical semantics** We saw that an adapted categorical semantics improved the semantics preservation on MNST→SVHN in Figure 3b. Here, we will finish the comparison of the effect of adapting the semantics categorical representation on accuracy for SVHN→MNIST and the FID for MNIST↔SVHN in Figure 8c

## C.3 ADDITIONAL RESULTS FOR SKETCH→REAL

We provide more results to support the results presented in Section 4.2 on the Sketch→Real task.

**Additional quantitative comparisons** We observe qualitatively in Table 4 that our method is lacking in terms of diversity with respect to the other methods that do not leverage any kind of semantics. This is not surprising because we penalize the network for generating samples that are unrealistic with respect to the semantics of the source sample.

**Effect of setting $\lambda_{sem} = 0$.** We demonstrated that not using the semantic loss considerably degraded the FID, in Table 3a. In Figure 9, we demonstrate qualitatively that the generated samples, when $\lambda_{sem} = 0$ suffers from the same problem as the baseline: the style is not conditional to the semantics of the source sample.

**Effect of the method to condition the semantics.** The method of conditioning the semantics in the network affects the generation, as observed in Table 3b. We present qualitative results in Figure 10 demonstrating the effect of not conditioning the semantics into any part of the translation network – while still using the semantic loss – and the effect of conditioning the style on the content

Table 4: Additional quantitative comparisons with the baselines. We qualitatively compare the baselines using all the Sketches→Reals categories using LPIPS (higher is better), NBD and JSD (lower is better).

| Data | CycleGAN | DRIT | EGSC-IT | StarGAN-V2 | CatS-UDT (ours) |
|------|----------|------|---------|------------|-----------------|
| LPIPS | 0.713 | **0.736** | 0.064 | 0.672 | 0.065 |
| NDB | 18 | 16 | 19 | 16 | **12** |
| JSD | 0.139 | **0.025** | 0.044 | 0.029 | 0.033 |

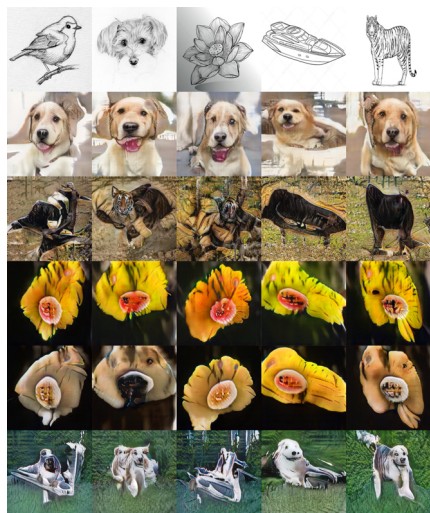

Figure 9: Sketch→Real using CatS-UDT with $\lambda_{\text{sem}} = 0$. Samples on the first row are the source samples. Samples on the subsequent rows are generated samples.

representation. In the latter case, we consider the semantics as categorical labels adapted to the sketches and the reals as well as semantics defined as the representation from a VGG network trained on classifying ImageNet.

In the first case, the network fails to generate diverse samples and essentially ignores the style input. We conjecture that this happens due to two reasons: (1) The content network and the generator cannot extract the semantics of the source image due to its constraints, relying on the style injected using AdaIN. (2) The mapping network generates the style unconditionally of the source samples; the style for one semantic category might not fit for another (e.g. the style of a tiger does not fit in the context of generating a speedboat). Therefore, to avoid generating, for example, a speedboat with the style of a tiger, the translation network ignores the mapping network.

In the second case, the network fails to generate samples like real images when using categorical semantics. We demonstrate such phenomenon in Figure 10b. The failure is similar to the one observed when the content encoder downsamples the source image beyond a certain spatial dimension. In both these cases, the generated samples lose the spatial coherence of the source image. Without the spatial

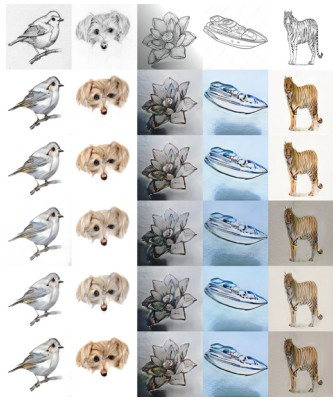

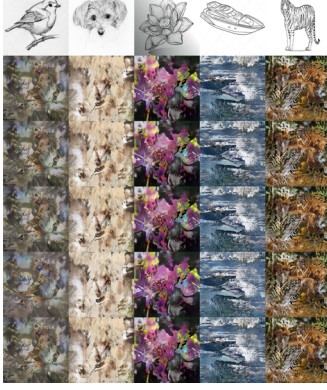

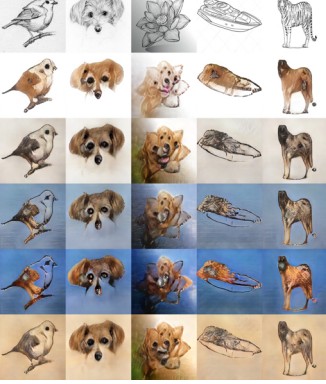

(a) Not conditioning the translation network.

(b) Condition the content representation with categorical semantics.

(c) Condition the content representation with VGG features.

Figure 10: Qualitative effect of the method to condition the semantics in the translation network in Sketches→Reals. Samples on the first row are the source samples. Samples on the subsequent rows are generated samples.

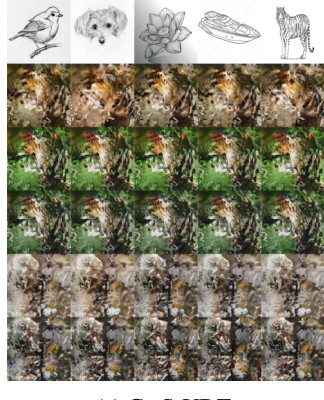

(a) CatS-UDT.

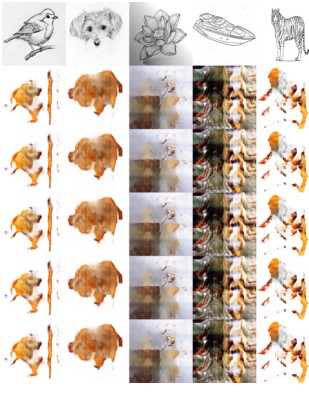

(b) CycleGAN.

Figure 11: Effect of the representation spatial dimension on the generation of Sketches→Reals. For (a) and (b), we downsample the content representation to a $4 \times 4$ feature map. Samples on the first row are the source samples. Samples on the subsequent rows are generated samples.

representation, the generator cannot leverage this information to facilitate the generation. Coupled with the fact that the architecture of the generator assumes access to such a spatial representation and the low number of samples, this explains why it fails at generating sensible samples. In this case, the spatial representation must be lost due to the addition of the categorical semantic representation and the semantic loss. We conjecture that by minimizing the semantic loss, the network tries to leverage the semantic information, interfering with the content representation. Furthermore, we tested a setup similar to the one presented in EGST-IT (Ma et al., 2019) where the semantics is defined as the features of a VGG network in Figure 10c. We see that this failure is not present in this case.

**Effect of the spatial dimension of the content representation.** We present examples of samples generated when the spatial dimension of the content representation is *too* small to preserve spatial coherence throughout the translation in Figure 11. In this example, we downsample until we reach a spatial representation of $4 \times 4$ for both our method and CycleGAN. We included CycleGAN to demonstrate that this effect is not a consequence of our method. In both cases, we see that the translation network fails to properly generate the samples as previously observed and discussed. This further highlights the importance of the inductive biases in these models.

**Additional generation for each classes.** We provide additional generations for each of the categories considered in Sketches→Reals in Figure 12 for more test source samples. In the fourth column of the dog panel in Figure 12b and the third column of the tiger panel in Figure 12e, we see a failure case of our method which can happen when a sketch gets mis-clustered. In the first case, the semantic network miscategorizes the dog for a tiger. In the second case, the semantic network miscategorizes the tiger for a dog. This further demonstrates the importance of a semantics network that categorizes the samples with high accuracy for the source and the target domain.

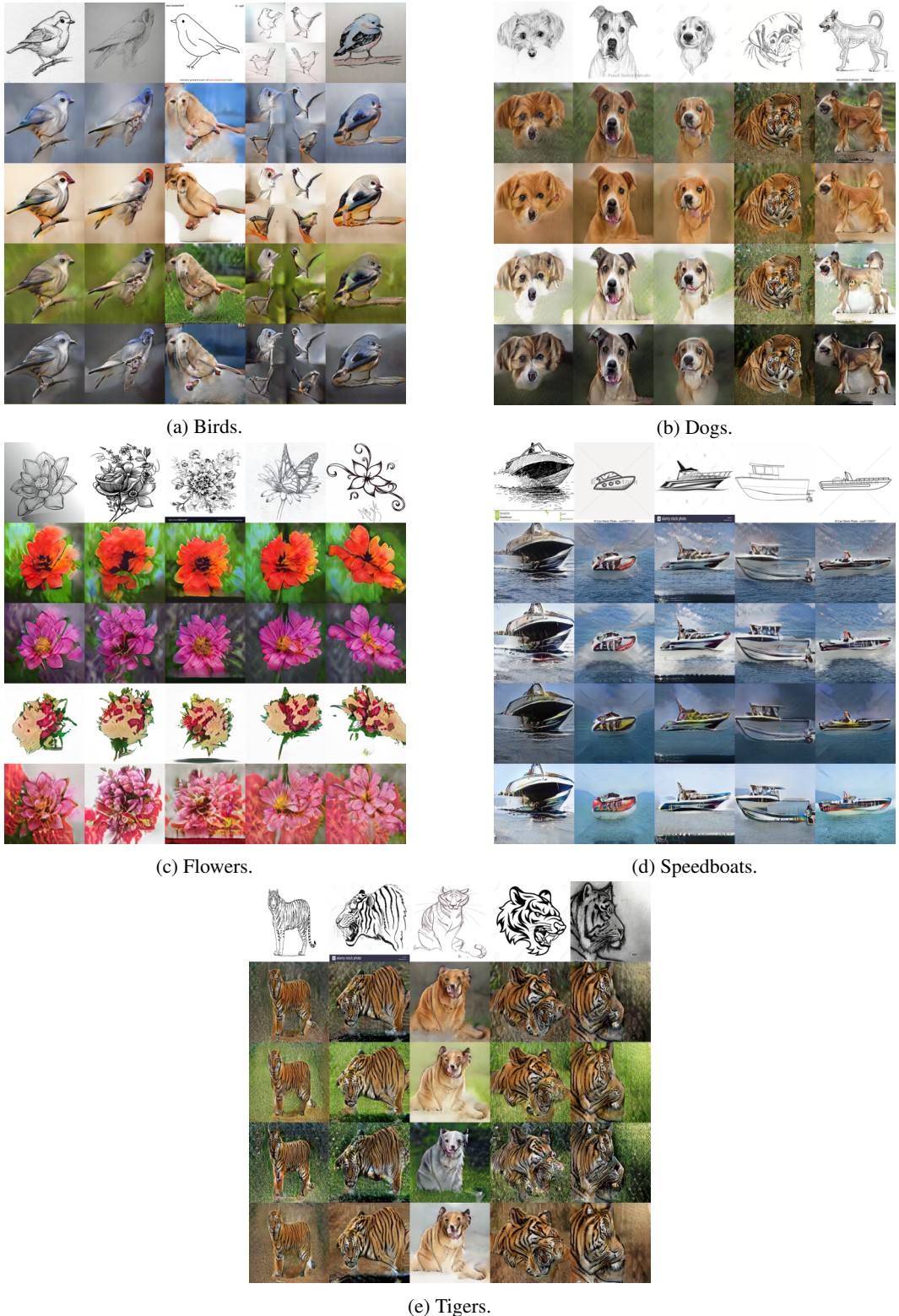

(a) Birds.

(b) Dogs.

(c) Flowers.

(d) Speedboats.

(e) Tigers.

Figure 12: Additional Sketches→Reals generations for each semantic categories.

