# OpenReview forum: "Integrating Categorical Semantics into Unsupervised Domain Translation"
_ICLR.cc/2021/Conference — ICLR 2021 Poster_

### Official Review · AnonReviewer2 · 2020-10-28
**Official Blind Review #2**

**Rating:** 7
**Confidence:** 2

**Review:**

This paper presents unsupervised domain translation (UDT), considering two scenarios: Semantic Preserving Unsupervised Domain Translation (SPUDT) and is Style-Heterogeneous Domain Translation (SHDT). This study uses MNIST and SVHN datasets for demonstrating SPUDT and Sketches and Reals samples from the DomainNet dataset for demonstrating SHDT. Although the method uses different components depending on the scenario (or dataset), the presented architecture is essentially the same.

The method consists of the framework for learning invariant categorical semantics across domains (Section 3.1) and semantic style modulation to make SHDT generations consistent (Section 3.2). Section 3.1 consists of unsupervised representation learning, clustering, and unsupervised domain adaptation. However, this section did not describe the first two components in detail probably because the method uses different components depending on the datasets (described in Section 4). Unsupervised domain adaptation is realized by the minimization problem described in P4. Section 3.2 describes content encoders, semantics encoders (obtained in Section 3.1), mapping networks, style encoders, and generator. The section presents the loss functions (adversarial loss, cycle-consistency loss, style-consistency loss, style diversity loss, and semantic loss).

Section 4 compares the proposed method with UDT baselines under SPUDT and SHDT scenarios. The experiments on MNIST and SVHN show that the proposed method achieved high accuracy in domain translation and high quality in the generation (measured by FID). The experimental results on Sketches and Reals show that the proposed method yields high-quality generations. Overall, the experiments demonstrate the importance of incorporating the semantics category in UDT.

PROS

It was interesting to see that the performance of UDT was drastically improved by incorporating semantic categories induced by the clustering algorithms.

The presented method was reasonable and well designed.

CONS

The main part of this paper is dense. I had to go back and forth between the main body and the appendix a lot.

This paper does not seem to emphasize the technical novelty of this work. Although Section 3.1 was designed to present a novel framework, it does not explain the detail of the components. Methods for unsupervised representation learning and clustering are explained in Section 4. The component of unsupervised domain adaptation may be a new proposal in this work. However, this paper did not explain this component in the main body; I had to read Appendix B.5 to understand the formula presented in P4.

The model architecture described in Section 3.2 is mostly based on previous work (e.g., adversarial loss, cycle-consistency loss, etc.). The semantic loss presented in this section may be a new addition to UDT. But again, this is not emphasized in Section 3.2.

QUESTIONS

P4: "We describe those regularizers in more detail in Appendix B.5."
If this part is a proposal in this paper, it should be explained as the main content of this paper. If I do not read Appendix B.5., I have no idea how domain 2 is involved in the minimization problem.

MINOR COMMENTS:

P1: "That are mappings biased toward the identity."
"That" -> "Those"

It may be convenient to have equation numbers so that reviewers/readers can locate a formula directly.

---

> ### Author Response · Authors · 2020-11-16
> **Response to Reviewer 2**
>
> We thank the reviewer for their review and appreciate the constructive comments.
>
> Our framework for learning domain invariant categorical semantics (Section 3.1) is agnostic to the instantiation of each constituent. Moreover, we believe that such instantiation should depend on the particular problem that a practitioner tackles (e.g the method for learning a representation may be different if the problem considers a dataset of images or a dataset of texts). For this reason, we defer the precise definition of the methods used in our experiments to the Appendix. That being said, we added a discussion around the concrete methods used in our work for each of the constituent in the text of Section 3.1 and added a more precise definition of the clustering of learned representations. Hopefully, these changes make the framework easier to understand.
>
> We have also put more emphasizes on the technical novelty in the body of the work by indicating what we propose in the text. Moreover, we have numbered the equations that are our contribution in the work because we agree that such numbering may be convenient when referencing an equation.

---

### Official Review · AnonReviewer3 · 2020-10-28
**An interesting work but lacking promising results**

**Rating:** 4
**Confidence:** 4

**Review:**

Summary: This paper proposes to learn the categorical semantic features in an unsupervised manner to enhance the ability of image translation at preserving high-level features, and obtains some good results on Semantic-Preserving Unsupervised Domain Translation and Style-Heterogeneous Domain Translation problems.


Major issues:
- The proposed method seems to be a combination of current works. The main contribution of this work may be leveraging the unsupervised representation learning for semantic features extraction.

- The quality of generated images is still not satisfactory with such rapid development of GANs.

- The experiment and qualitative evaluation are too limited. Only two image translation tasks are conducted for comparison, and little visual results are given. It will be preferred if some common I2I tasks results are given. Only FID is used, adding other metrics, such as LPIPS, NDB, and JSD, will be more convincing.


Minor issues
- What are the essential differences between SPUDT and SHDT problem? How does the model solve the two problems according to their differences?

---

> ### Author Response · Authors · 2020-11-16
> **Response to Reviewer 3 1/2**
>
> We thank the reviewer for their review and are eager to engage with you regarding your concerns about our paper. We address each of the issues that you presented.
>
> ### The proposed method seems to be a combination of current works. The main contribution of this work may be leveraging the unsupervised representation learning for semantic features extraction.
>
> The main thesis of our work is that unsupervised extraction of domain-invariant categorical semantics enables applications in UDT that were not previously possible (e.g. SPUDT and SHDT, discussed in the Introduction). To our knowledge, we are the first to illustrate a working UDT pipeline that succeeds at the SPUDT and SHDT tasks, demonstrating the relevance of domain-invariant categorical semantics. We also demonstrate through a baseline study that the current SOTA methods for UDT do not perform well at those tasks. While we are in the broadest sense "leveraging unsupervised representation learning for semantic features extraction", our overall development of a working solution to existing problems in UDT involves several novel technicalities in both the extraction of semantics itself and and the integration of semantics into UDT (both described in Section 3). We empirically demonstrate the relevance of these contributions through comparison and ablation studies in Section 4 (e.g. comparison of our method for extracting semantic features vs. traditional methods in Figure 3.d), ablation of using the semantic loss in UDT in Figure 3.a) and Table 3.a) or comparison of our method for integrating the semantics vs. traditional methods in Table 3.b).) We believe that showcasing such possibilities can also inspire similar applications in other cross-domain unsupervised problems where semantic information is necessary but not readily available through supervision.
>
> ### The quality of generated images is still not satisfactory with such rapid development of GANs.
>
> We used the same architecture of the state-of-the-art method at the time of writing the article which in turn provides a more fair comparison to the baselines. Moreover, the core problem that we tackle in this paper is orthogonal to the generation quality. In other words, we believe that the results we present demonstrates the problems with SPUDT and SHDT. We also believe that the generation quality is sufficient to demonstrate that incorporating semantics resolves these problems.
>
> ### The experiment and qualitative evaluation are too limited. Only two image translation tasks are conducted for comparison, and little visual results are given. It will be preferred if some common I2I tasks results are given. Only FID is used, adding other metrics, such as LPIPS, NDB, and JSD, will be more convincing.
>
> We picked two images datasets that we believed meaningfully highlighted two issues un-tackled by the UDT literature (SPUDT and SHDT). Our experiments focused on demonstrating the issues with the current UDT baselines and how our contributions really tackle and solve those issues. Another image dataset tackling any of the same issues would be redundant, leaving less space for discussion and empirical demonstration. The common I2I tasks are designed to not have the issues that we are concerned with in our paper. For these reasons, we argue that the number of translation tasks that we demonstrate is sufficient.
>
> We provide extra qualitative results in Figure 6 and 12 for MNIST-SVHN and Sketches→Reals respectively, in Appendix C.
>
> We added a table with the metrics that you recommended in Appendix C.3 (Table 4). LPIPS quantitatively demonstrates what we can observe in Figure 4: our method produces samples that are less diverse than the baselines. This is expected considering that our method penalizes the model for generating samples that are un-realistic with respect to the semantics of the source sample. This reduces the span of possible generation given a source sample, therefore reducing the diversity.

---

> ### Author Response · Authors · 2020-11-16
> **Response to Reviewer 3 2/2**
>
>
> ### What are the essential differences between SPUDT and SHDT problem? How does the model solve the two problems according to their differences?
>
> Both the tasks of Semantic-Preserving Unsupervised Domain Translation (SPUDT) and Style Heterogeneous Domain Translation (SHDT) are described in Section 1.
>
> In our experiments, we noticed that if a target distribution was "easy" to generate (e.g. MNIST or SVHN), then every UDT baselines that we tried discarded any information from the source sample and simply generated the target distribution.
>
> On the other hand, if a target distribution was "hard" to generate (e.g. Real images) and had multiple semantic categories with distinct styles, then the UDT baselines that we tried were able to preserve the pose of the source sample, but failed at generating a style that is consistent with the semantics of the source sample.
>
> Therefore, the two problems are slightly different. On one case, the objective is to learn a translation that preserves the shared semantics. In the other case, the objective is to generate a style that is consistent with respect to the semantics of the source sample. But, these two problems share a central component: they require knowledge about the shared semantics. Our model solve this problem by incorporating semantics into the model and in the objective function used to train the translation model.

---

### Official Review · AnonReviewer1 · 2020-10-29
**An interesting and original idea**

**Rating:** 7
**Confidence:** 4

**Review:**

The paper addresses the domain translation problem and proposes a novel approach to translate images between domains in an unsupervised manner, by integrating unsupervised learning of domain-invariant semantic features between the two domains. The paper is well-written with a clear standing point and motivation, along with well-described contributions. The paper has an inclusive and sound theoretical comparison to related work. Evaluation is well-designed and includes previous work in the same context.
In overall, it is a good paper with an original and potentially inspiring idea and a convincing application of conditional GANs, would be an interesting read for many in the conference.

Typos
Page 1:
showing that such a translation
two situations where unsupervised domain translation
Page 7:
the representations of the real images

Comments
1. 'unsupervised domain translation methods work due to an inductive bias toward identity' in sections 1 and 3.2: need citation to support this statement
2. Considering clustering is one of the important parts of the contributions it may need more emphasis in the paper, it would be nice to restructure this part, extend the discussion of alternatives and an analysis to compare different approaches (also by moving some part of the discussion to the paper from the appendices).

---

> ### Author Response · Authors · 2020-11-16
> **Response to Reviewer 1**
>
> We thank the reviewer for their review of the paper and the good words. We are pleased to read that you found this paper potentially inspiring.
>
> Concerning the comment on 'unsupervised domain translation methods work due to an inductive bias toward identity', we corrected the citation in the updated revision. We replaced the wording with the one used in Galanti et al. characterizing the translation in UDT as being a mapping with 'minimum complexity'.
>
> Considering your second point, we added more discussions on unsupervised representation learning and clustering in Section 3.1. We put a clearer description of the clustering in our framework. We also discuss potential instantiations for each constituent that we considered in our work.

---

### Official Review · AnonReviewer4 · 2020-11-04
**Nice Results using Domain Invariant Categorical Semantics to Improve UDT**

**Rating:** 7
**Confidence:** 3

**Review:**

Summary: The authors use Domain Invariant Categorical Semantics to improve unsupervised domain translation (UDT). They learn these semantics in an unsupervised manner. They show how this can improve results on Semantic Preserving Unsupervised Domain Translation and Style Heterogeneous Domain Translation by doing experiments on MNIST<->SVHN (features traditionally learned are very different but digit identity could be the same) and Sketches->Reals (distinct styles) respectively.

---

Strengths:
- The visual results on both tasks/datasets are striking.
- The paper is simple to read and the idea is intuitive.
- Experiments are extensive, including an ablation on the losses and comparison against baselines.

Weaknesses:
- More examples of Sketches->Real could have been shown;
- When does this method fail? Some failure cases would be good

---

> ### Author Response · Authors · 2020-11-16
> **Response to Reviewer 4**
>
> We thank the reviewer for their review and the good words on the paper. We are glad to read that you appreciated the experiments and the clarity of the paper.
>
> We present more examples of Sketches→Reals in Figure 12 (Appendix). In sub-figure 12.e), third column, we demonstrate an example where our method fails. In this example, our model stylizes the sketch of a tiger with the style of a dog. This is caused by the semantic network inferring the wrong cluster for the source sample. We have added a discussion of this failure case in Section 5 of the revised version of our paper.

---

### Decision · Program_Chairs · 2021-01-07
**Final Decision**

**Decision:**

Accept (Poster)

**Comment:**

This paper studies the problem of unsupervised domain translation. Here translation does not refer to language translation. Instead, it refers to the idea of transferring high-level semantic features. Specifically, the authors look at digit style transfer (between MNIST/postal address numbers and SVHN/street view house numbers) and Sketches to Reals. The visuals look very convincing and the empirical results are strong, too. There is one weaker review but the authors address the concerns in their response and the reviewer did unfortunately not respond despite promting.